# Exploiting Scale Invariance and Rotation Equivariance for Sparse and Dense Artery Orientation Estimation

**Dieuwertje Alblas**[1]                                      D.ALBLAS@UTWENTE.NL
**Iris N. Vos**[2]                                           I.N.VOS-6@UMCUTRECHT.NL
**Julian Suk**[1]                                            J.M.SUK@UTWENTE.NL
**Christoph Brune**[1]                                       C.BRUNE@UTWENTE.NL
**Kak Khee Yeung**[3]                                        K.YEUNG@AMSTERDAMUMC.NL
**Birgitta K. Velthuis**[4]                                  B.K.VELTHUIS@UMCUTRECHT.NL
**Hugo J. Kuijf**[2]                                         H.KUIJF@UMCUTRECHT.NL
**Ynte M. Ruigrok**[5]                                       IJ.M.RUIGROK@UMCUTRECHT.NL
**Jelmer M. Wolterink**[1]                                   J.M.WOLTERINK@UTWENTE.NL

[1] *Dept. of Applied Mathematics, Technical Medical Centre, University of Twente, The Netherlands*

[2] *Image Sciences Institute, University Medical Center Utrecht, The Netherlands*

[3] *Amsterdam UMC, Vrije Universiteit Amsterdam, Dept. of Surgery, Amsterdam, The Netherlands*

[4] *Department of Radiology, University Medical Center Utrecht, The Netherlands*

[5] *Department of Neurology and Neurosurgery, University Medical Center Utrecht, The Netherlands*

## Abstract

We present SIRE, a modular estimator of local artery orientations that is Scale Invariant and Rotation Equivariant. These symmetries are preserved by operating on *spherical* image patches at multiple scales in parallel, and allow generalisation to arteries of unseen sizes and orientations. We embed SIRE into two different artery centerline tracking algorithms: a sparse, iterative tracker starting at a single seed point and a dense image filter serving as a cost function for connecting two bifurcation points. We show that SIRE can be used to obtain centerlines of arteries of various sizes and tortuosities by including datasets containing abdominal aortic aneurysms, coronary arteries and intracranial arteries.

**Keywords:** artery orientation, symmetry preservation, graph neural network

## 1. Introduction

Local estimation of artery orientation is an important step in many (semi-)automatic centerline extraction methods (Lesage et al., 2009). Previously proposed methods used a convolutional neural network (CNN) operating on a cubical image patch to estimate the local artery orientation and embedded this in various tracking algorithms (Wolterink et al., 2019; Su et al., 2023; Salahuddin et al., 2021). However, CNNs are not robust against variations in artery size and tortuosity, as the ratio between the image patch size and the artery diameter varies and CNNs are not equivariant to rotations. Therefore, these methods require retraining when tracking arteries in a different region of the human body. We present a scale-invariant, rotation-equivariant (SIRE) estimator for local artery orientation that generalises to vessels of unseen sizes and orientations due to its symmetry preservations. SIRE can be used to sparsely estimate vessel orientations at arbitrary image locations or on a dense Cartesian grid as an orientation filter. Here, we demonstrate both strategies

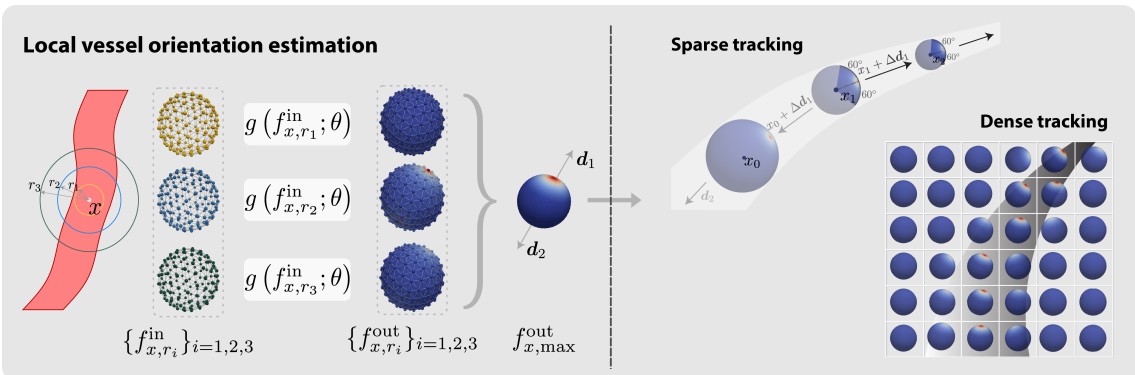

Figure 1: *Left:* SIRE overview. A graph neural network $g(\cdot; \theta)$ operates on multi-scale spherical image features *in parallel*, outputs are transformed into $f_{x,\max}^{\text{out}}$ by a maximum operation, from which local artery orientations are inferred. *Right:* Sparse and dense artery centerline tracking methods embedding SIRE.

in two common approaches to artery centerline extraction: iterative tracking from a single seed point, and minimum cost path extraction between two points. We demonstrate the generalisation of SIRE by including datasets containing coronary arteries, abdominal aortic aneurysms (AAAs) and intracranial arteries.

## 2. Materials & Methods

SIRE leverages scale- and rotational symmetries to obtain the local artery orientation at $x \in \mathbb{R}^3$ based on image data (Alblas et al., 2023). SIRE consists of a *single* gauge-equivariant mesh convolutional neural network (GEM-CNN) $g(\cdot, \theta)$ (De Haan et al., 2021), operating in parallel on multiple, nested spherical image patches centered around $x : f_{x,r_i}^{\text{in}}$ (Fig 1, *left*). For each user-defined scale $r_i$, $g(\cdot, \theta)$ outputs a scalar field $f_{x,r_i}^{\text{out}}$ on the spherical surface. Scale invariance is obtained by aggregating the scale-wise responses into an output $f_{x,\max}^{\text{out}}$ using a permutation-invariant maximum operation. The locations of local maxima of $f_{x,\max}^{\text{out}}$ correspond to the local artery orientations at $x$. As orientation estimation using SIRE is intrinsic to the sphere, it rotates along with the artery orientation.

SIRE can be used to estimate the local artery orientation sparsely on a single point, or densely on a Cartesian grid as an image filter. For both strategies, we implemented an automatic centerline extraction algorithm (Fig. 1, *right*). The first algorithm is a sparse iterative tracker initiated at a single seed point traversing the centerline with a predefined step size $\Delta = 0.5$ mm and terminates once a stopping criterion is met. This stopping criterion is based on an uncertainty measure of $f_{x,\max}^{\text{out}}$. In the second algorithm, we applied SIRE densely as an image processing filter. We constructed a cost function based on the mean cosine similarity between the artery orientations predicted by $g(\cdot, \theta)$ on neighbouring vertices that we used to connect two bifurcation points using Dijkstra's algorithm.

For sparse tracking, we used two datasets containing arteries with different diameters: the ASOCA dataset (Gharleghi et al., 2023) with 40 contrast-enhanced CT scans of coronary

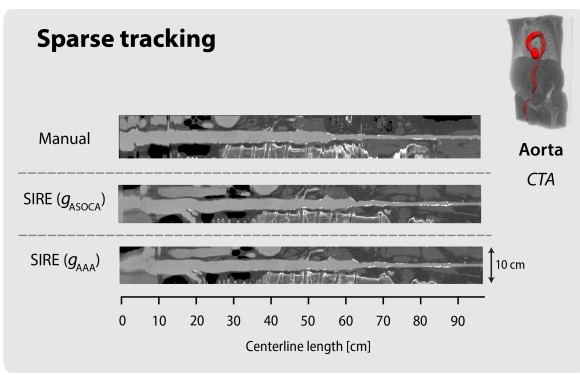 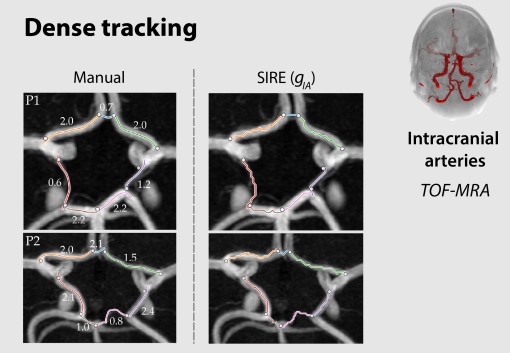

Figure 2: *Left*: automatic aorta centerlines obtained using the sparse tracking algorithm using $g_{\text{ASOCA}}$ and $g_{\text{AAA}}$. *Right*: intracranial artery centerlines obtained by connecting bifurcation points using dense tracking, with artery diameters in mms.

arteries (1-7 mm) and an in-house dataset with 108 contrast-enhanced CT scans of AAAs, (2-11 cm). The dense tracking algorithm was applied to another in-house dataset containing 32 3D time-of-flight magnetic resonance images of the intracranial arteries (0.5-4.6 mm).

To quantitatively assess the extracted centerlines, we use the average inside (AI) distance for the sparse tracking algorithm (Schaap et al., 2009). We used the Fréchet distance to evaluate centerlines obtained using the dense tracking algorithm.

## 3. Experiments & Results

We used the sparse tracking algorithm to track aortas in the AAA dataset starting at a manually placed seed point in the aorta at the renal bifurcation. We compared $g_{\text{AAA}}$ to $g_{\text{ASOCA}}$, i.e. $g(\cdot, \theta)$ trained on the AAA and ASOCA data, respectively. Fig. 2, *left* shows multi-planar reconstructions of the AAA centerlines extracted using $g_{\text{AAA}}$ and $g_{\text{ASOCA}}$. The AI-distances were $2.23 \pm 0.75$ mm and $2.52 \pm 0.91$ for $g_{\text{AAA}}$ and $g_{\text{ASOCA}}$, respectively.

We used the dense centerline tracking algorithm to obtain centerlines of the intracranial arteries, featuring $g(\cdot, \theta)$ trained on the intracranial artery dataset ($g_{\text{IA}}$). Fig. 2, *right* shows maximum intensity projections (MIP) and the extracted intracranial artery centerlines for two patients. The median Fréchet distance for all centerlines was 2.18 (1.33) mm.

## 4. Discussion & Conclusion

We have presented SIRE: a Scale Invariant and Rotation Equivariant, modular estimator for local artery orientations, that we embedded in two different centerline tracking algorithms. Both tracking algorithms resulted in accurate centerlines for arteries of varying size ($<$ 1 mm to $>$ 7 cm) and tortuosities. Moreover, by training SIRE on coronary arteries and evaluating on aortas, we demonstrated the generalisation of SIRE without the need for additional retraining.

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
