# OpenReview forum: "Exploiting Scale Invariance and Rotation Equivariance for Sparse and Dense Artery Orientation Estimation"
_MIDL.io/2024/Short_Papers — MIDL 2024 Short Papers_

### Official Review · Reviewer_o6R2 · 2024-04-25

**Confidence:** 3
**Final Rating:** 3.5

**Review:**

This paper introduces a scale invariant and rotation equivariant (SIRE) estimator of local artery orientations to extract centerlines of arteries of various sizes and tortuosities.

The proposed idea is interesting and of great interest to the MIDL community. However, the experimental results in Fig. 2 show that while the centerline predicted by the proposed method shares similarity with the manual results, it is notably less smooth than the ground truth. Additionally, the experiments lack comparison with other existing approaches, making it difficult to justify the advantages of the model.

---

### Decision · Program_Chairs · 2024-04-26

Accept